# Combination of a Self-Regulation Module and Mobile Application to Enhance Treatment Outcome for Patients with Acne

**DOI:** 10.3390/medicina56060276

**Published:** 2020-06-04

**Authors:** Yi-Shan Liu, Nan-Han Lu, Po-Chuen Shieh, Cheuk-Kwan Sun

**Affiliations:** 1Department of Pharmacy, Tajen University, Pingtung 907391, Taiwan; aliceliutw@yahoo.com.tw; 2School of Chinese Medicine for Post-Baccalaureate, College of Medicine, I-Shou University & Department of Dermatology, E-Da Hospital, Kaohsiung 824410, Taiwan; 3School of Medicine, College of Medicine, I-Shou University & Department of Radiology, E-Da Hospital, Kaohsiung 824410, Taiwan; draliceliu@yahoo.com.tw; 4School of Medicine, College of Medicine, I-Shou University & Department of Emergency Medicine, E-Da Hospital, Kaohsiung 824410, Taiwan; aliceliutw@gmail.com

**Keywords:** acne vulgaris, acne assessment, acne self-regulation inventory, mobile application, live chat bot

## Abstract

*Background and Objectives:* Acne, an inflammatory disorder of the pilosebaceous unit associated with both physiological and psychological morbidities, should be considered a chronic disease. The application of self-regulation theory and therapeutic patient education has been widely utilized in different health-related areas to help patient with a chronic disease to attain better behavioral modification. The present study aims at investigating the treatment efficacy of combining a self-regulation-based patient education module with mobile application in acne patients. *Materials and Methods:* This was one-grouped pretest–posttest design at a single tertiary referral center with the enrollment of 30 subjects diagnosed with acne vulgaris. Relevant information was collected before (week 0) and after (week 4) treatment in the present study, including the Acne Self-Regulation Inventory (ASRI), Cardiff Acne Disability Index (CADI), and Dermatology Life Quality Index (DLQI) that involved a questionnaire-based subjective evaluation of the patient’s ability in self-regulation and quality of life as well as clinical Acne Grading Scores (AGS) that objectively assessed changes in disease severity. To reinforce availability and feasibility, an individualized platform was accessible through mobile devices for real-time problem solving between hospital visits. *Results*: Thirty subjects completed the designed experiment. An analysis of the differences between scores of pretest and posttest of ASRI demonstrated substantial elevations (*p* < 0.001). The questionnaire survey of CADI and DLQI dropped significantly after the application of a self-regulation-based patient education module with a mobile application, revealing substantial reductions in both parameters (*p* < 0.001). The sign test demonstrated a remarkably significant difference in AGS (Z = −7.38, *p* < 0.001), indicating notable improvement in the clinical severity of acne after treatment. *Conclusions*: After incorporating modern mobile application, a self-regulation-based therapeutic patient education module could significantly improve treatment outcomes among acne patients.

## 1. Introduction

Like other chronic diseases, acne has recently been categorized as a chronic [1] inflammatory disorder [2,3] of the pilosebaceous unit [4] associated with both physiological [5] and psychological morbidities [6,7,8].

Many studies have reported successful therapeutic experience through combining medical treatment with a well-designed patient education program [9]. Self-regulation theory (SRT) [10], which proposes that human behavior is the outcome of interactions among the individual, environment, and behavior [11], has been widely applied in various health-related topics [12,13] to improve patient adherence and attain behavioral modifications [14,15]. Self-regulated learning, which embodies the ingredients of goal-setting and self-efficacy [16], plays a pivotal role in the exercise of personal agency via its strong impact on emotion, thought, motivation, and action [17]. Moreover, therapeutic patient education (TPE) helps patients obtain the skills to handle their life with a chronic disease [18,19]. TPE, which is a patient-centered process involving the transfer of skills from trained professionals to patients, is a four-step process: (1) making an educational diagnosis to understand patients’ insight; (2) setting appropriate educational objectives; (3) helping patients to acquire skills; and (4) assessing success of the program [20]. On the other hand, a mobile phone is an easily accessible tool that possesses multiple functions and has become indispensable in people’s daily lives [21].

Through utilizing a mobile application incorporated with SRT and TPE, the present study attempted to build a therapeutic module that would benefit acne patients as well as dermatologists and nurse practitioners who care for this patient population. It focused on 4-week follow-ups of acne outpatients whose treatment course can easily be traced and analyzed by their mobile devices. To reinforce availability and feasibility, an individualized platform was accessible for real-time problem solving by a live chat bot through the mobile application.

## 2. Materials and Methods

### 2.1. Study Participants

From December 2018 to December 2019, all participants visiting a single tertiary referral center (Department of Dermatology of E-Da Hospital, Taiwan) with a confirmed diagnosis of acne made by qualified dermatologists were enrolled. All patients had attended scheduled study follow-up visits at the dermatology outpatient clinic and signed informed consent forms for participation in the study.

The inclusion criteria were: (a) ≥14 years old; and (b) diagnosed with acne by qualified dermatologists of E-Da Hospital. The exclusion criteria were: (a) illiterate; (b) diagnosis of systemic diseases, endocrinologic and psychiatric disorders; (c) pregnant; (d) subjects taking isotretinoin or systemic corticosteroids within 30 days; (e) those taking systemic medications likely to aggravate or abate acne such as oral phenytoin or any other epileptic, finasteride, spironolactone, or flutamide, or testosterone; (f) individuals receiving facial procedures such as chemical peeling, laser, microdermabrasion, or phototherapy within 30 days.

### 2.2. Study Materials and Procedures

The current study adopted a one-grouped pretest–posttest design at a single medical care center with the enrollment of 30 subjects diagnosed with acne vulgaris. The study duration was 4 weeks with visits at week 0 (baseline) and week 4. Relevant information was collected before (week 0) and after (week 4) treatment, including an objective assessment of acne severity by dermatologists with the Acne Grading Scores (AGS) as well as patients’ subjective evaluation of their self-adjustment ability using the Acne Self-Regulation Inventory (ASRI) and the negative impacts on their life quality using the Cardiff Acne Disability Index (CADI) and the Dermatology Life Quality Index (DLQI).

The self-regulation-based therapeutic patient education module was a four-step process (Figure 1). The aim of the first step was to educate the acne patients to understand the nature and pathogenesis of acne. In the clinic, the dermatologist diagnosed and graded the disease and informed the patient of the appropriate application of the prescribed topical drug. The patient was then introduced to another room to receive an individual session about acne by a nurse practitioner. Patient education was done by the nurse practitioner and included, firstly, making patients understand the nature and pathogenesis of acne by using education brochures. In the second step, the nurse practitioner taught the patients the skills to better manage their disease, according to their difficulties and resources. The skills included medication use, skin care, diet habits, and lifestyle modification to decrease the severity of acne. We integrated this knowledge during individual session to create a take-home message for patients (Figure 2) as a constant reminder by mobile phone. The third step was about the acquisition of skills by the patients. During this step, we used an ASRI to help them to reinforce their ability to self-regulate to meet their therapeutic objectives and to promote better adherence to the treatment. The fourth step concerned the assessment of TPE, the efficacy of which was evaluated with AGS, ASRI, DLQI, and CADI.

Additionally, during the 4 weeks before the final visit, a Dermatological Live Chat Bot (DLCB) [22] was built for acne patients through their mobile phones to enhance the therapeutic efficacy of shared decision-making (SDM) [23] via unlimited access to advice on disease management. The questionnaires were completed and archived through a Google Form by patients’ mobile phones. Patients could ask and discuss acne therapy-related questions as well as find brochures about acne therapy through the interface of Facebook messenger through their mobile phones without time or space restrictions (Figure 3). The interface of the DLCB was constructed through words, with the core technology including cloud natural language process and natural language understanding based on artificial intelligence. The address of uniform resource locator (URL) is https://mobile.facebook.com/profile.php? id = 2329997470545842, titled Acne Expert, accessed from 01 December 2018. The study protocol was approved by the institutional review board (IRB) of E-Da Hospital and conducted in compliance with the Declaration of Helsinki (IRB approval no.: EMRP09108N from 01 December 2018).

### 2.3. Study Parameters

(a)Acne Grading Scores (AGS): A very simple grading of clinical acne severity was described in the 2003 report from the Global Alliance for better outcomes of acne treatment [24]. This basic systematics, which was designed to be used in a routine clinic, aims at mapping treatment plan by clinical presentations. Acne severity is graded from none (0), mild (1), moderate (2), to severe (3).(b)Acne Self-Regulation Inventory (ASRI): ASRI comprises in total 31 items based on four subscales of the self-regulation theory, including self-monitoring (6 items), a self-judgment subscale (6 items), a self-reaction subscale (12 items), and self-motivation (7 items). A 4-point Likert scale was used to assess the subjects’ responses to their understanding of the disease and treatment method as well their compliance to the proposed therapeutic strategy [25]. The scoring system was as follows: 3–very strong; 2–strong; 1–moderate; 0–no response. The higher the total ASRI scores, the better the self-regulation ability and compliance of acne patients under treatment.(c)Cardiff Acne Disability Index (CADI): The index is a brief and simple 5-item questionnaire designed to assess the disability caused by acne regarding the negative emotional and social impacts of acne on patients’ daily lives. Hence, the questionnaire is a patient’s subjective assessment of acne-induced disability, and provides relevant information to dermatologists [26]. The higher the CADI scores, the larger the impact of disability caused by acne.(d)Dermatology Life Quality Index (DLQI): The index is the most widely used quality of life instrument for patient self-evaluation of dermatological problems. The testing subjects are asked to choose the most appropriate answers to ten questions involving the emotional, social, and physical aspects affecting their quality of life by the disease [27,28]. The total score of the DLQI is proportional to the negative effects from the skin disease.

### 2.4. Statistical Analysis

All statistical analyses were executed using the SPSS software (Version 17.0, SPSS Inc., Chicago, IL, USA). Paired-sample t tests and sign tests were applied to evaluate the significance of difference between pretest and posttest scores for the continuous (ASRI, CADI, and DLQI) and categorical (AGS) variables, respectively. A *p* < 0.05 was considered statistically significant.

## 3. Results

### 3.1. Demography of Study Subjects

The thirty participants mentioned in Section 2.2 fitting the inclusion and exclusion criteria (Section 2.1) were recruited for assessing the efficacy of combining a mobile application with a self-regulation-based therapeutic patient education module in the treatment of patients with acne vulgaris. The study population consisted of 21 females and 9 males with a mean age of 28.45 ± 6.27 (range, 14–40), with the pretest acne severity by a mean Acne Grading Scores of 2.25 ± 0.31.

### 3.2. Acne Self-Regulation Inventory (ASRI)

The analysis of the differences between pretest and posttest ASRI scores in the subjects demonstrated substantial elevations after the implementation of the self-regulation-based TPE module combined with a mobile application for acne patients (all *p* < 0.001) (Table 1), suggesting highly significant improvements in self-regulation capability during the course of treatment.

### 3.3. Acne-Associated Disability and Quality of Life

The evaluation of the acne-associated disability and quality of life was performed with CADI and DLQI, respectively. The analysis of the differences between the pretest and posttest scores in the testing patients revealed substantial reductions in both parameters after treatment (*p* < 0.001) (Table 1), suggesting significant improvements in disability and quality of life after treatment.

### 3.4. Acne Grading Scores (AGS)

The comparison of AGS before and after the application of self-regulation based TPE module assisted with mobile device for acne patients showed no increase in score for all patients. While decreases were noted in the majority of 27 patients, there was no change in scores in the other three patients. The sign test demonstrated a remarkably significant difference (Z = −7.38, *p* < 0.001) (Table 2), indicating notable alleviation of the clinical severity of acne after treatment.

## 4. Discussion

Acne, a pilosebaceous inflammatory disorder [4] related to both physiological and psychological morbidities [4,5,6,7], should now be considered to be and treated as a chronic disease [1,2,3]. Because of the complex properties of acne as a chronic disease rather than a self-limiting disorder, dermatologists should act as both healers and educators. Further, acne patients should understand the chronic nature of acne that may involve frequent flare-ups and remissions. However, to date, there is no evidence-based acne treatment approach integrating the concept of SRT and the process of TPE. The present study, which aimed at investigating the therapeutic effectiveness of combining a mobile application with a self-regulation-based therapeutic patient education module for patients with acne, demonstrated not only the importance of the educational role of a dermatology care team in acne treatment, but also the efficacy of combining a self-regulation-based therapeutic patient education module with a mobile application for enhancing both subjective and objective therapeutic outcomes.

Because the concept of SRT suggests that human behaviors are the result of interactions between the individual, environment and behavior, the module of the present study was developed in three dimensions: (1) dermatologists for acne outpatient assessment; (2) nurse practitioners for the clarification of educational objectives; and (3) patients for the completion of objective questionnaires. The results of the current study showed that combining the three-dimensional interventional approach with an internet live chat bot platform that not only provided unrestricted patient access to acne treatment information but also allowed real-time feedback from the patients could significantly reinforce treatment efficacy. Furthermore, the mobile device-assisted self-regulation-based therapeutic patient education module could be applied to other typically incurable chronic skin diseases, such as atopic dermatitis or psoriasis vulgaris, taking into consideration its time-effectiveness and the importance of minimizing social contacts to reinforce the policy of social distancing during the period of a viral pandemic (e.g., the coronavirus disease of 2020). 

There are several limitations to this study. Firstly, the number of study subjects was relatively small. Secondly, all patients were enrolled from a single medical institute; therefore, the results may not be globally applicable. However, the results demonstrated significant improvements, highlighting the therapeutic effectiveness of combining a self-regulation module with a mobile application for patients with acne in both physical and psychological aspects; not only could it alleviate the clinical severity of acne but it could also enhance the patients’ self-regulation capability as well as improve their acne-induced disability and their quality of life.

## 5. Conclusions

It is necessary for dermatologists to educate their patients that acne is a chronic disease instead of a self-limiting disorder. By utilizing an internet platform on mobile devices, the implementation of an effective self-regulation-based therapeutic patient education module for acne patients could be a time- and cost-effective measure that enables constant modifications of education and therapeutic strategies through real-time bilateral interactions to meet therapeutic goals without the restrictions of time and space.

## Figures and Tables

**Figure 1 medicina-56-00276-f001:**
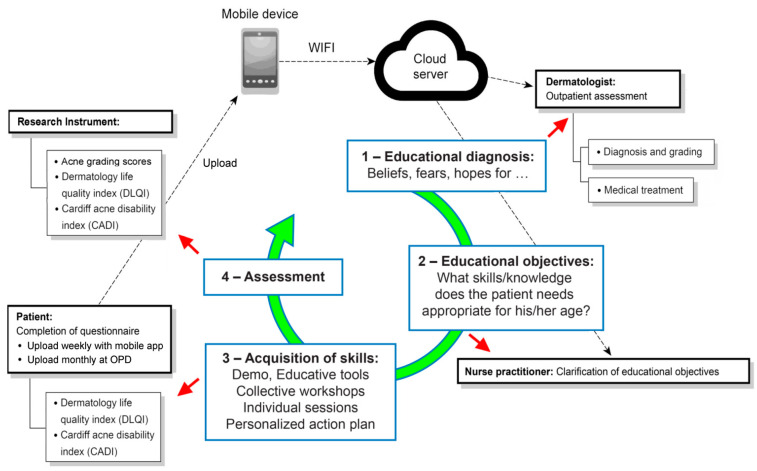
The study module, which was developed in four steps: (**1**) dermatologists for acne outpatient assessment; (**2**) nurse practitioners for the clarification of educational objectives; (**3**) patients for the completion of questionnaires for objective assessment; and (**4**) during the period between hospital visits, patients’ immediate inquiry or feedback could be uploaded to a cloud server through an internet live chat bot platform using their mobile devices without time or space limitations. OPD: Out Patient Department.

**Figure 2 medicina-56-00276-f002:**
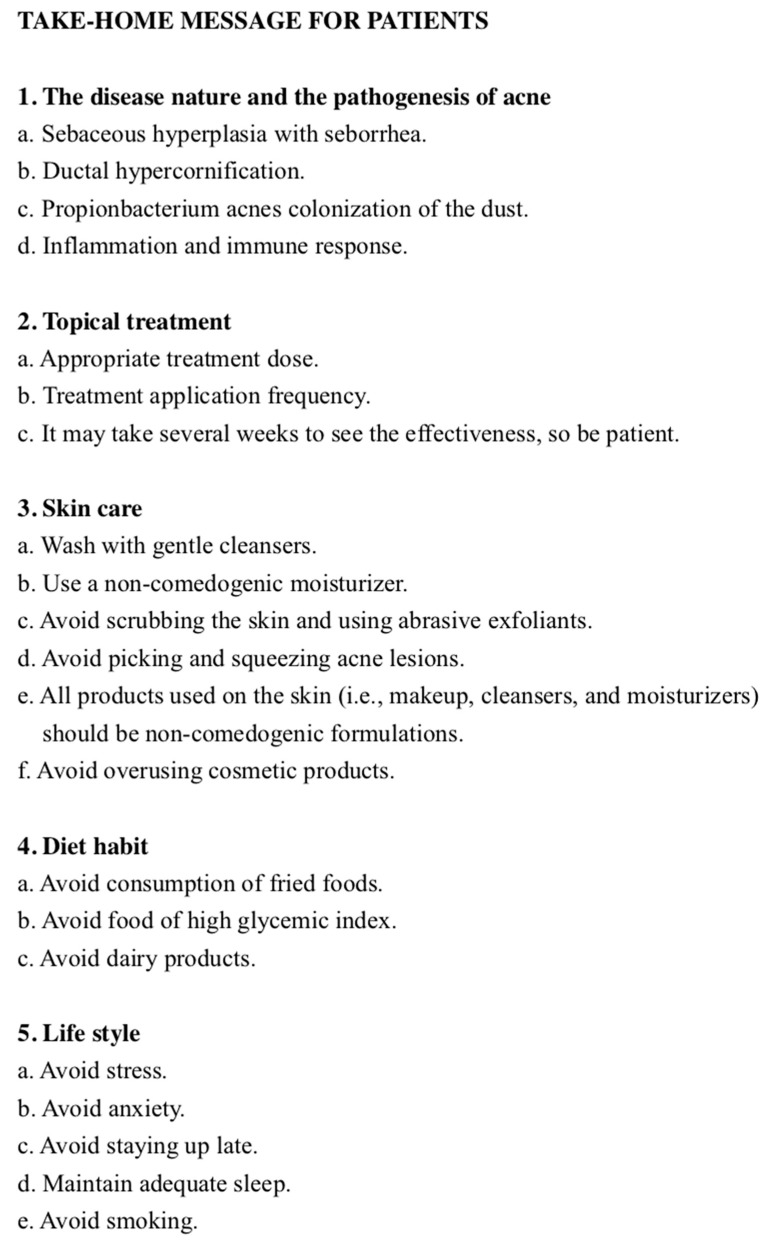
Take-home message for patients.

**Figure 3 medicina-56-00276-f003:**
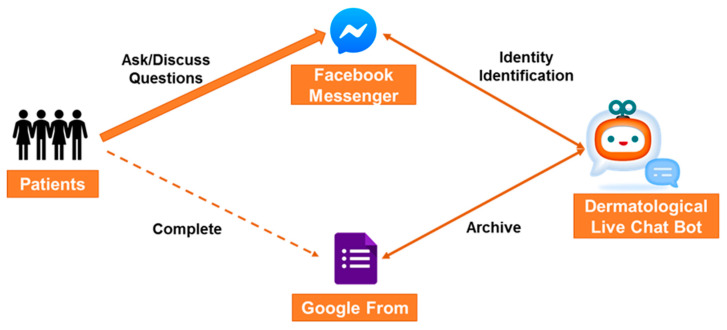
A Dermatological Live Chat Bot (DCL) platform was built for acne patients through their mobile phones to enhance the effect of shared decision-making.

**Table 1 medicina-56-00276-t001:** Comparison of the mean scores between the pretest and posttest Acne Self-Regulation Inventory (ASRI) for the evaluation of self-regulation capability and the Cardiff Acne Disability Index (CADI) and Dermatology Life Quality Index (DLQI) for the assessment of acne-associated disability and the quality of life of acne patients (*n =* 30).

	Pretest Score (Mean ± SD)	Posttest Score (Mean ± SD)	t	*p*-Value *
**ASRI**	56.09 ± 16.15	70.07 ± 16.12	9.23	<0.001
**CADI**	7.45 ± 3.21	4.15 ± 2.39	−9.59	<0.001
**DLQI**	11.34 ± 5.88	4.70 ± 3.43	−9.74	<0.001

ASRI: Acne Self-Regulation Inventory; CADI: Cardiff Acne Disability Index; DLQI: Dermatology Life Quality Index. * Significance of differences determined by paired *t*-test.

**Table 2 medicina-56-00276-t002:** Sign test statistics, pretests and posttests, of acne grading scores (AGS) for the evaluation of clinical acne severity.

	(Posttest–Pretest) Scores
Z	−7.38
Asymptotic Significance (2-Tailed)	0.000 *

AGS: Acne Grading Scores. * Significance of differences determined by the sign test (*p* < 0.001).

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
