# Peer review of "Combination of a Self-Regulation Module and Mobile Application to Enhance Treatment Outcome for Patients with Acne"

_medicina, 2020, doi:10.3390/medicina56060276_

Round 1

Reviewer 1 Report

The authors have addressed all my previous comments

Author Response

  I really appreciated that you approved our manuscript. The authors have addressed all my previous comments. We hope that more readers to get benefit from it to help their acne patients clinically.

Reviewer 2 Report

Thank you for your revision. The quality of the manuscript has improved significantly.

However, please include more information about the study cohort as previously asked. (age, gender, acne severity) It would be interesting to see, whether younger patients who might be more familiar with social media/apps could benefit more than older patients.

Author Response

Point 1: Please include more information about the study cohort as previously asked. (age, gender, acne severity) It would be interesting to see, whether younger patients who might be more familiar with social media/apps could benefit more than older patients.

Response 1:

  I really appreciated that you approved the quality of our manuscript. Thank for your kindly reminder. The data you suggested was added in the section 3.1.

This manuscript is a resubmission of an earlier submission. The following is a list of the peer review reports and author responses from that submission.

Round 1

Reviewer 1 Report

Your paper covers an interesting topic however please address/change the following issues:

  1. Where did the study take place?
  2. Please provide information about the study cohort (demographic data, acne severity, current treatment, past medical history)
  3. Once an abbreviation has been introduced, please stay consistent and use it throughout the whole manuscript and do not reintroduce it. (e.g Self-regulation theory (SRT))
  4. Please explain in more detail which topics were covered in the mobile application . (page 2, line 83-85) Only therapeutic guidance or information such as skin care, nutrition, pathogenesis? Maybe provide further figures so that the audience can better understand the mobile application. 
  5. How can the improvement of the clinical severity of acne be explained? (page 4, line 145) Please include the treatment of patients or is this supposed to be due to the mobile application?
  6. Please discuss your results in the discussion rather than restating information which were already included in the introduction.

Reviewer 2 Report

I really enjoyed reading the manuscript presented by Liu et al. One of the weaknesses I observed was that the actual application and central parts of the assessment such as the ASRI could only be understood when also reading the 2016 paper. I would encourage the authors to use more the supplementary material in order to give the reader a more complete understanding without having to pull multiple papers together or weaken the conciseness of the current manuscript.

Further I would feel that it is necessary to provide somewhere a copy or access of the app used in the study. I understand that there might be language limitations but it would help others to use similar approaches. Especially as an AI trained software will never be the same if reproduced a second time.

It would have been desirable to include in the study a comparison group which does not use the app or a kind of sham app in order to assess the magnitude of the placebo effect. However I understand that this is not a trivial task and is not possible in retrospectively. I would still feel that the current findings are highly relevant as the limitation is obvious to the skilled reader.